# A Silicon Nanowire Array Biosensor Fabricated by Complementary Metal Oxide Semiconductor Technique for Highly Sensitive and Selective Detection of Serum Carcinoembryonic Antigen

**DOI:** 10.3390/mi10110764

**Published:** 2019-11-11

**Authors:** Xun Yang, Yun Fan, Zhenhua Wu, Chaoran Liu

**Affiliations:** 1School of Electronic and Information Engineering, Foshan University, Foshan 528000, China; sophiafan69@126.com; 2State Key Laboratory of Transducer Technology, Shanghai Institute of Microsystem and Information Technology, Chinese Academy of Sciences, Shanghai 200050, China; 3College of Electronics and Information, Hangzhou Dianzi University, Hangzhou 310018, China; liucr@hdu.edu.cn

**Keywords:** carcinoembryonic antigen, silicon nanowire array, highly sensitive, highly selective, low cost

## Abstract

In this paper, we present a highly sensitive and selective detection of serum carcinoembryonic antigen (CEA) based on silicon nanowire (SiNW) array device. With the help of traditional microfabrication technology, low-cost and highly controllable SiNW array devices were fabricated. After a series of surface modification processes, SiNW array biosensors show rapid and reliable response to CEA; the detection limit of serum CEA was 10 fg/mL, the current signal is linear with the logarithm of serum CEA concentration in the range of 10 fg/mL to 100 pg/mL. In this work, SiNW array biosensors can obtain strong signal and high signal-to-noise ratio; these advantages can reduce the production cost of the SiNW-based system and promote the application of SiNWs in the field of tumor marker detection.

## 1. Introduction

Tumor markers can reflect the occurrence and development of cancer; they exist in tissues and body fluids of cancer patients and can be detected by immunological, biological and chemical methods [1,2,3,4,5,6,7,8]. As an important tumor marker, the reliable detection of carcinoembryonic antigen (CEA) plays a key role in the diagnosis and treatment of various cancers [9,10]. In the blood of healthy adults, the concentration of CEA is generally less than 5 ng/mL. Malignant tumors can lead to a significant increase in CEA concentration; when it continues to rise 5–10 times, there is a possibility of intestinal cancer. The concentration of CEA is closely related to the clinical stage and surgical outcome of cancers. So, dynamic monitoring of CEA concentration has important clinical significance in the diagnosis and treatment of colorectal cancer, gastric cancer, lung cancer, breast cancer and other malignant tumors [11,12,13,14,15]. 

With the development of immunology and molecular biotechnology, we can use various biological characteristics of CEA to improve the diagnostic rate of tumors. The clinical detection methods of CEA include electrochemical immunoassay, chemiluminescence immunoassay, fluorescence immunoassay, piezoelectric immunoassay and surface plasmon resonance [16,17,18,19,20]. These methods have some shortcomings, such as having low sensitivity, being very time-consuming and requiring expensive equipment, so high sensitivity and low-cost detection of CEA is still challenging and is in its infancy. With the maturity of micro-nanofabrication technology, the detection technology of CEA will gradually develop to the direction of miniaturization and integration. A large number of studies have been reported on the detection of CEA by quantum dots, graphene, silicon nanowires (SiNW) and other nanomaterials [21,22,23,24,25,26,27,28,29], because of the high surface-to-volume ratio, nanomaterials have been widely used to detect biospecific recognition reaction between antigen and antibody. The biosensor based on quantum dots has the advantages of high sensitivity, high selectivity and high reliability, but it needs to use fluorescent labels to read out the signal; at the same time, it needs precise optical equipment to convert into readable signal, which makes the detection process too complex to achieve low-cost and real-time detection of CEA concentration [30,31]. The biosensor based on graphene can directly convert CEA concentration values into readable electrical signal, which can realize rapid and real-time detection of CEA; however, as the graphene prepared by the existing synthetic methods is difficult to transfer, the fabrication process is complex [32]. These shortcomings limit the further application of quantum dots and graphene in the detection of tumor markers. Therefore, it is necessary to develop a highly sensitive, label-free, low-cost and real-time CEA detection method. As a novel one-dimensional material, a large number of studies have shown that SiNWs have the potential to meet these performance requirements [33]. At the same time, compared with a single SiNW device, the SiNW array device can obtain stronger signal and higher signal-to-noise ratio [34,35]. However, the large-scale application of SiNWs is limited by the lack of low-cost fabrication technology of SiNW arrays. The fabrication methods of SiNW arrays include chemical vapor deposition (CVD) [36,37], metal catalytic chemical etching (MACE) [38,39,40,41,42,43] and top-down fabrication based on nanofabrication technology. Compared with the first two preparation methods, top-down fabrication technology has advantages in controllability and reproducibility. However, most of the top-down fabrication technologies need to use the expensive lithography equipment [44,45,46]. Therefore, it is urgent to develop a low-cost SiNW array preparation technology. In a previous report, we presented a wafer-level and highly controllable top-down fabrication technology for SiNW array devices [47], where the entire fabrication process is compatible with the complementary metal oxide semiconductor (CMOS) technology and only uses traditional microfabrication technology, such as lithography, reactive ion etching, and wet etching. Because of the high surface-to-volume ratio, the detection limit of the anti-CEA-modified SiNW biosensor was 10 fg/mL; the current signal is linear with the logarithm of serum CEA concentration in the range of 10 fg/mL to 100 pg/mL. In this report, the biosensor has the characteristics of low cost, good repeatability, high sensitivity and being label-free, and so is suitable for high performance real-time detection of CEA.

## 2. Experimental Details

### 2.1. Fabrication of Silicon Nanowires (SiNW) Array Device

The fabrication process for SiNW array device is schematically shown in Figure 1. First, a 100-nm-thick layer of silicon nitride (Si_3_N_4_) was deposited on a (111) silicon-on-insulator (SOI) wafer. Then, Si_3_N_4_ and the top silicon of SOI is etched to oxide layer. Based on the anisotropic wet-etching, highly downscaled silicon walls can be formed by the corrosion of potassium hydroxide; after the self-limiting oxidation, SiNWs are generated on the top center of each wall. Source electrode, drain electrode, grid electrode and trench isolation were fabricated in specific locations. After removing the silica wall, the SiNW array device was finally fabricated. The scanning electron microscope (SEM) photograph of the SiNW array device is shown in Figure 1h; 120 SiNWs are regularly distributed in the comb structure, which are protected by the top silicon nitride film. The device was installed on a printed circuit board (PCB), and the electrical connection between the device and PCB is realized by gold wire bonding; the photo of the packaged SiNW array device is shown in Figure 1i.

### 2.2. Materials

3-aminopropyltriethoxysilane (APTES), glutaraldehyde, ethanol amine, prostate specific antigen (PSA) and bovine serum albumin (BSA) were purchased from Sigma-Aldrich (St. Louis, MI, USA), CEA and CEA antibody were purchased by Fitzgerald Inc (Elkader, IA, USA), CEA serum standards were obtained from Bio-Rad (Hercules, CA, USA). All chemicals were used carefully in accordance with the manufacturer’s instructions, and all protein solutions were diluted to the desired concentrations with phosphate buffer saline (PBS).

### 2.3. Surface Modification of SiNW Array

Through a series of chemical reactions, CEA antibodies can be assembled on the surface of SiNW arrays and the modified SiNW array device can be used to identify and detect CEA. The specific modification path is shown in Figure 2. Firstly, the surface of SiNW arrays was cleaned by oxygen plasma with power of 70 W. At the same time, a large number of hydroxyl groups were formed on the surface of SiNWs to make them more hydrophilic, so as to facilitate the subsequent modification process. The device was immersed in 2% APTES solution overnight, amino groups were modified on the surface of SiNW arrays, and then 10% glutaraldehyde solution was dripped on the device. Glutaraldehyde was chemically bonded with amino groups on the surface of SiNW arrays to form a self-assembled monolayer terminated with aldehyde groups. CEA antibody solution was added, which bound to aldehyde groups on the surface of SiNWs. Finally, ethanol amine solution was added, which bound to unreacted glutaraldehyde to reduce non-specific binding between external disruptor and the surface of SiNW array.

## 3. Results and Discussion

### 3.1. Sensitivity of the SiNW Array Biosensor

We have previously reported the relation between the dimensions of SiNW and its sensitivity [47]:(1)$$\mathrm{sensitivity}=\frac{\Delta I}{I_{0}}=\frac{\left(I-I_{0}\right)}{I_{0}}=\frac{K}{\bar{w}}-\frac{K}{{\bar{w}}^{2}}(w-\bar{w})$$
where *I_0_* and *I* are the current signals before and after the binding of CEA, respectively, *K* is a constant, $\bar{w}$ is the average dimension of SiNWs, and *w* is the dimension of the SiNW. From Equation (1), it can be concluded that only when the dimensions of the SiNWs are almost equal can the sensitivity of each SiNW be similar, which also means that each SiNW has the same signal *ΔI*. The signal of each SiNW can be superimposed, so SiNW array device can obtain a stronger signal than a single SiNW device. As shown in Figure 3a,b, the signal intensity of the 120 SiNW array device is about 100 times higher than that of a single SiNW device. In a previous report, we have proposed that the signal-to-noise ratio of the device can be improved by increasing the number of SiNWs [47]. The effect can be described by the following equation:(2)$$SNR=\frac{NS}{n}$$
where *SNR* is the signal-to-noise ratio, *N* is the number of SiNWs, *S* is the signal of a single SiNW, and *n* is the random noise of the SiNW-array device. As shown in Figure 3a,b, the signal-to-noise ratio has been greatly improved. So, integrated fabrication of SiNWs can achieve stronger signal and higher signal-to-noise ratio; these advantages can reduce the production cost of the SiNW-based system and promote the application of SiNWs.

The combination of CEA and CEA antibodies will lead to sharp increase in charge density on the surface of SiNWs. Just as gate voltage affects field effect transistors, the changed surface potential will affect carrier concentration in SiNWs, leading to a detectable change of the conductance of a biosensor. As shown in Figure 3c,d, a control experiment was conducted to confirm the specific binding of antigen and antibody on the surface of SiNWs. When 100 fg/mL CEA solution was added to the surface of unmodified SiNW arrays, there was no obvious current change, which indicates that there was no non-specific binding between CEA and SiNWs (Figure 3c). Significantly, when CEA solution was added to the surface of an anti-CEA-modified SiNW array biosensor, the current signal rose rapidly; because of the high surface-to-volume ratio, the minimum detection limit of the SiNW array biosensor was 1 fg/mL for CEA. In this work, we use p-type devices to detect CEA concentration, in which the majority carrier is the hole. When negatively charged CEA is specifically adsorbed on the surface of SiNWs, the concentration of holes in SiNWs will be greatly increased, resulting in a decrease in resistance and an increase in current signal. As shown in Figure 3d, we measured the signals of the biosensor in CEA solutions of different concentration. The real-time current of SiNWs ascended in a stepwise manner with change of CEA concentration from 1 fg/mL to 10 pg/mL. The series of 10^−15^, 10^−14^, 10^−13^, 10^−12^ and 10^−11^ g/mL solutions of CEA resulted in current increases of 36%, 83%, 150%, 191% and 234%, respectively. In order to achieve accurate quantitative analysis, we have drawn the standard curve of the biosensor in Figure 3d, which shows the relationship between the current signal and the CEA concentration. It can be seen that the logarithm of the measured liquid concentration is strictly proportional to the current signal, which lays a foundation for accurate measurement of CEA concentration in the future.

### 3.2. Specificity of the SiNW Array Biosensor

Figure 4 shows the specificity of SiNW array biosensor which was modified with CEA antibody. We used two kinds of non-homologous proteins to conduct real-time control experiments with CEA. When 10 mg/mL BSA was introduced onto the surface of SiNWs, the current signal hardly changed, and then the current increased rapidly after adding 1 fg/mL CEA. The relative change of current signal was 43%, indicating that it is difficult for BSA to adsorb on the surface of SiNWs even if the concentration of BSA solution is very high. Similarly, when 100 μg/mL PSA solution is introduced onto the surface of SiNWs, the current change of SiNW biosensor is not obvious. The above results show that the anti-CEA-modified SiNW biosensor has no specificity for BSA and PSA but has good specificity for CEA; excellent specificity can greatly improve the diagnostic rate in the detection of tumor markers.

### 3.3. CEA Detection in Real Sample 

The types and concentrations of cancer markers in real samples are important references for early detection and prognosis evaluation of cancer patients. Here, we used the SiNW array biosensor to quantitatively detect CEA concentration in real samples. Firstly, CEA serum samples were diluted to different concentrations with 0.01 M PBS buffer solution. As shown in Figure 5, a stable baseline was obtained by adding PBS buffer solution to the surface of SiNW arrays, the current signal has risen rapidly after dripping CEA serum solution, and we found that the sensitivity of SiNW array biosensor was 10 fg/mL for CEA serum solution. We measured the signals of the biosensor in CEA serum solution of different concentrations, and the real-time current of SiNWs ascended in a stepwise manner with the change of CEA concentration from 10 fg/mL to 100 pg/mL; 10^−14^, 10^−13^, 10^−12^, 10^−11^ and 10^−10^ g/mL solutions of CEA resulted in a current increases of 74%, 137%, 192%, 243% and 318%, respectively. The SiNW array biosensor can get a readable electrical signal directly, which realizes rapid and real-time detection of serum CEA solution; the biosensor can reach the steady-state in less than 150 s. In order to achieve accurate quantitative analysis, we also have drawn the standard curve of the biosensor (Figure 5b). The high sensitivity detection of real samples demonstrates the potential of the biosensor in clinical application.

The traditional detection technology of cancer markers has some disadvantages, such as having low sensitivity, being time-consuming and requiring expensive detection equipment. Table 1 shows the minimum detection limit and test range of our biosensors and the traditional detection devices; from this table, we can see that by using the traditional detection methods it is still difficult to achieve high sensitivity detection of CEA. Although the detection technologies based on graphene and quantum dots can achieve high sensitivity detection, they also have some shortcomings, such as complex device preparation processes and numerous detection procedures, which limit their further application. In this work, CEA detection technology based on the SiNW array device can effectively remedy these shortcomings. This technology is label-free and electrically readable, having the further advantages of high sensitivity, low cost, real-time detection and so on. It has broad application prospects for cancer diagnosis. Compared with other nano-material devices, SiNW array devices can obtain stronger signal with higher signal-to-noise ratio. These advantages make it easy to integrate with low-cost electronic detection circuits, which makes it possible to build low-cost electronic detection systems for cancer markers.

## 4. Conclusions

In conclusion, we have developed a highly sensitive and selective method for the detection of serum CEA by using the SiNW array device. The SiNW arrays with low cost and high controllability were fabricated by traditional micro-fabrication technology. The whole fabrication process only adopts the traditional micro-nano fabrication technology, which is compatible with CMOS technology. Compared with electron beam lithography and deep ultraviolet lithography, the technology can fabricate cheaper devices and is suitable for mass production. The biosensor has fast and reliable response to CEA, it can detect serum CEA solution as low as 10 fg/mL quickly and sensitively. Because of their advantages of being label-free and having high sensitivity, high selectivity and low cost, the SiNW array biosensors are expected to provide new solutions in the field of cancer marker detection in the future.

## Figures and Tables

**Figure 1 micromachines-10-00764-f001:**
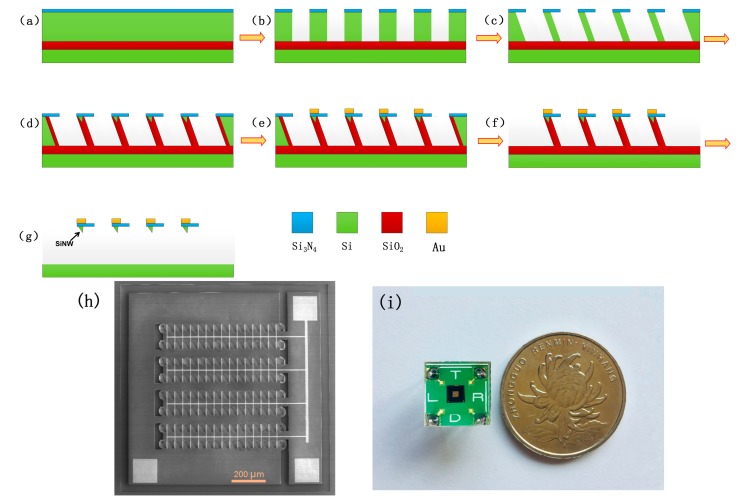
The fabrication process of the silicon nanowires (SiNW) array device. (**a**) The silicon nitride was deposited on a (111) silicon-on-insulator (SOI) wafer. (**b**) The reaction ion etching to form etched cavities. (**c**) The anisotropic wet-etching to form silicon walls between the adjacent etched cavities. (**d**) The self-limiting oxidation to form SiNWs. (**e**) The fabrication of source electrode, drain electrode and top-grid electrode. (**f**) The fabrication of the trench isolation. (**g**) Removing the silica walls. (**h**) The scanning electron microscope (SEM) photograph of the SiNW array device. (**i**) The photo of the packaged SiNW array device.

**Figure 2 micromachines-10-00764-f002:**
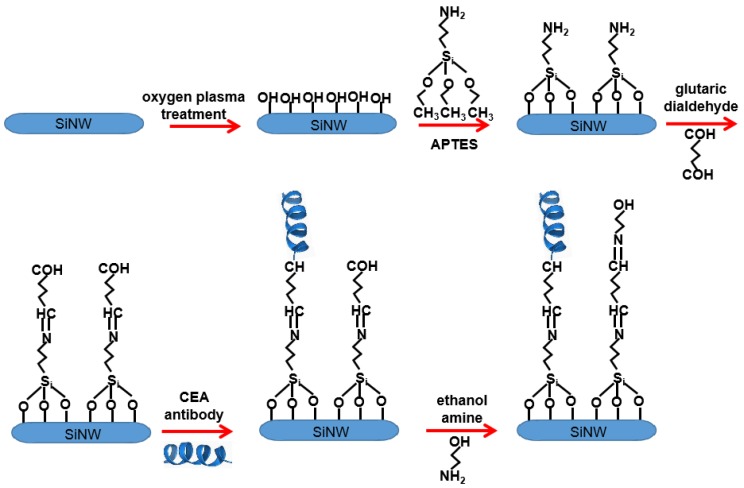
The process of surface modification of the SiNW array.

**Figure 3 micromachines-10-00764-f003:**
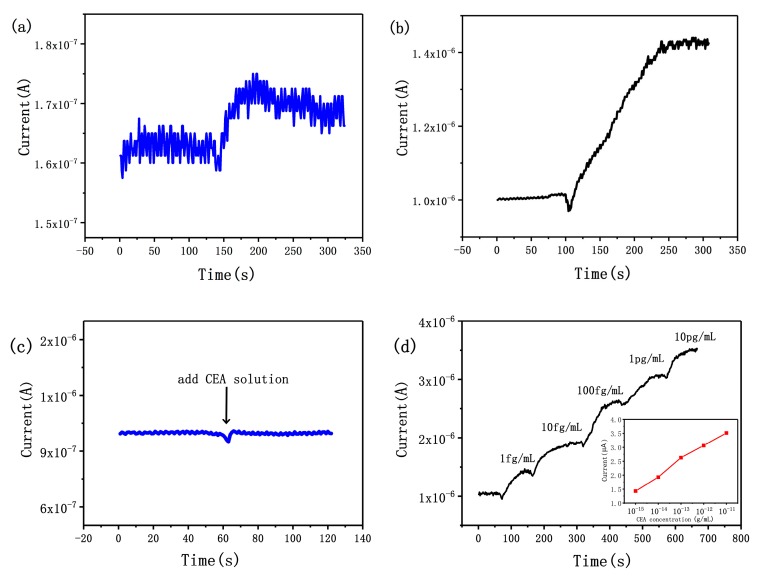
(**a**) Time course of current for single SiNW biosensor when 1 fg/mL carcinoembryonic antigen (CEA) solution was added. (**b**) Time course of current for SiNW array device when 1 fg/mL CEA solution was added. (**c**) Time course of current for unmodified SiNW array when 100 fg/mL CEA solution was added. (**d**) The sensor response to different CEA concentrations.

**Figure 4 micromachines-10-00764-f004:**
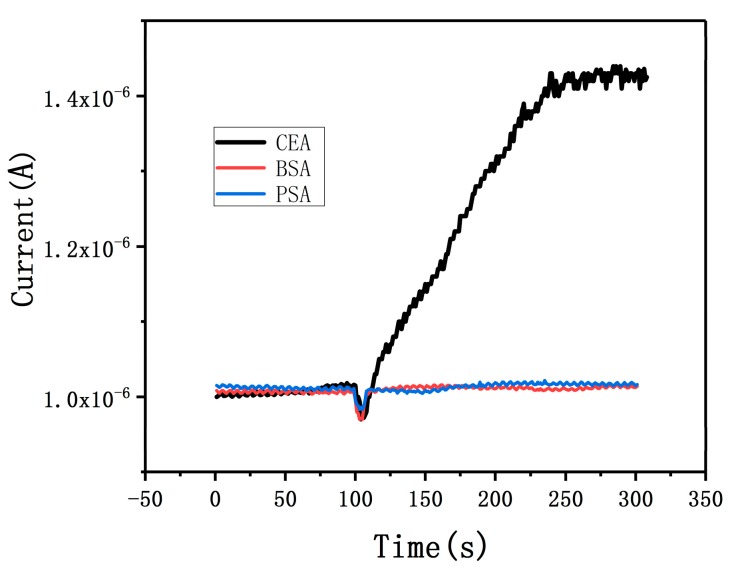
Characterization of the specificity of the anti-CEA-modified SiNW biosensor, displaying the time course of current for 10 mg/mL of BSA, 100 μg/mL of PSA and 1 fg/mL of CEA.

**Figure 5 micromachines-10-00764-f005:**
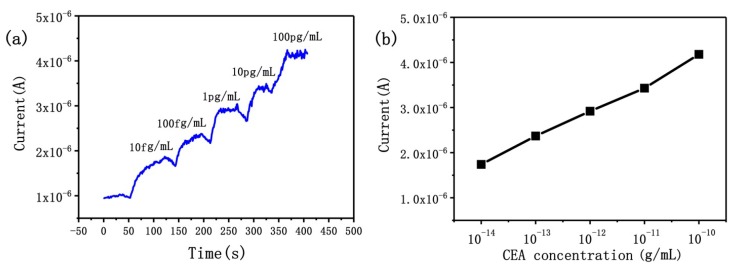
(**a**) The sensor response to different serum CEA concentrations. (**b**) The current signal versus exponential increases of serum CEA concentration.

**Table 1 micromachines-10-00764-t001:** Comparison of detection limit and test range.

Methods	Detection Limit (pg/mL)	Test Range (pg/mL)	References
Electrochemical immunoassay	500	500–50,000	[16]
Chemiluminescence immunoassay	610	610–250,000	[17]
Fluorescence immunoassay	210	210–200,000	[18]
Piezoelectric immunoassay	66,700	66,700–466,700	[19]
Surface plasmon resonance	3000	3000–400,000	[20]
SiNW array	0.001	0.001–10	This work

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
