# Peer review of "A Silicon Nanowire Array Biosensor Fabricated by Complementary Metal Oxide Semiconductor Technique for Highly Sensitive and Selective Detection of Serum Carcinoembryonic Antigen"

_micromachines, 2019, doi:10.3390/mi10110764_

Round 1

Reviewer 1 Report

In this article, the authors introduced a new fabrication technique for SiNW array device with the ultimate goal of quantifying CEA in bodily fluids with enough specificity and selectivity. The article is very well written with low to none issues noted by this reviewer. This reviewer is impressed with the 10fg/ml selectivity that was demonstrated via experiments. This reviewer also significantly appreciated the thorough specificity study presented.

The only comment this reviewer has that needs addressing is the fact that why do we need to introduce a sensor with 10fg/ml sensitivity to measure CEA concentrations of 5ng/ml in healthy samples and 50ng/ml with malignant tumors.  It seems like overkill. 

Author Response

Point-to-point responses to the reviewers’ comments

We appreciate the reviewers’ meticulous reading and valuable suggestions on our manuscript. Those comments are all valuable and very helpful for revising and improving our paper, as well as the important guiding significance to our researches. We have studied comments carefully and have made correction which we hope meet with approval. The point-to-point responses to the reviewers’ comments are as flowing.

Reviewers' comments:

Reviewer #1:

In this article, the authors introduced a new fabrication technique for SiNW array device with the ultimate goal of quantifying CEA in bodily fluids with enough specificity and selectivity. The article is very well written with low to none issues noted by this reviewer. This reviewer is impressed with the 10fg/ml selectivity that was demonstrated via experiments. This reviewer also significantly appreciated the thorough specificity study presented.

The only comment this reviewer has that needs addressing is the fact that why do we need to introduce a sensor with 10fg/ml sensitivity to measure CEA concentrations of 5ng/ml in healthy samples and 50ng/ml with malignant tumors.  It seems like overkill. 

Response: We thank you for appreciating our efforts in highly sensitive and selective detection of serum carcinoembryonic antigen. Because there are a lot of interferences in human blood samples, the concentration of CEA in blood can not be directly measured by nano materials at present. The CEA samples we purchased have been purified and diluted, and its concentration is far lower than that of normal human blood. Therefore, we think that high sensitivity detection of low concentration CEA still has its application value.

Reviewer 2 Report

I would like to recommend the authors to deeply revise their manuscript based on : 

1- the English requires professional editing 

2- Abstract should be re-written with specific results and achievements, not general sentences, specially like the first sentence. 

3- The title should be re-written, the mention method is not novel at all and also authors mentioned lots of similar reports in references section. 

4- CMOS-MEMs technique in the title should be mentioned with long name

5- Introduction section is very poorly written and there is no obvious trend and strange claims on that. 

6- the sensing method is not mentioned in the Materials and method section. 

7- the characterization methods is not mentioned

8- to confirm the formation of SiNWs and immobilization of the antibodies on the surface, some characterization methods like FTIR, XPS, SEM, and etc. are needed

9- the Authors have to show the blocking step of the non-reacted carboxyl groups with ethanol amine in Figure 2. 

10- how authors could claim that the glutaraldehyde did not cross-linked two neighbour amine groups during immersion of electrode in the glutaraldehyde solution? it can block the electrode surface 

Author Response

Dear reviewer,

Thank you for your insightful comments, which are significant to improve our article after the corresponding revision. I send you the revised manuscript and the Point-to-point responses in the form of attachment.

Reviewer 3 Report

The authors report CEA detection technology by a CMOS technique with SiNW array. Such matured CMOS technology is useful to highly sensitive devices without expensive equipment. The SiNW array device based on CMOS technology has a potential application to the biological and medicinal chemistry and medical front. Therefore, the manuscript can be suitable for Micromachines after appropriate modification.

[p.2, L. 44] The authors state many CEA sensors were low-sensitive and long time-consuming. However, the authors did not compare the properties of their SiNW array device with the previously reported devices. Therefore, the authors should discuss it in the manuscript. [p.2, L. 86] The abbreviation SOI requires the full-spelling. [p.4, L. 123] Glutaraldehyde has no carboxyl group. Why? [p.5, L. 158] How was the minimum detection sensitivity determined? The sensitivity of 1 fg/mL is not likely to the detection limit. And again, is the 1 fg/mL sensitivity superior to the previously reported devices? [Equation (1)] The equation (1) was just introduced but not discussed regarding this manuscript. No variables, K, w and w-bar were determined in the SiNW array there. [Figure 3(d)] Why is the current proportional to the logarithm of the concentration of the CEA? [Figure 3(a) and (b)] Time constant of the SiNW array was slower than that of the single SiNW device as in Figure 3(a) and (b). The device of the SiNW array required ~150 sec to reach the steady-state and that of the single SiNW did ~50 sec. Is the response time reproducible?  Furthermore, again, is the response time shorter than the other detection technologies of CEA which the authors referred to in the manuscript? [p.6, L. 186] Why the SiNW device is specific for CEA, not BSA and PSA? [p.6, L. 192] What is the definition of the “real sample”? What is the difference between the real sample from the CEA solution in the previous section? [p.6, L. 196] The unit is missing. 0.01 PBS is 0.01 M PBS? [p.7, L. 227] The authors conclude “it can detect serum CEA solution as low as 10 fg/mL quickly and sensitively”. But the detection speed was not discussed in the manuscript.  

Author Response

(The authors gave the same response as above.)

Round 2

Reviewer 2 Report

Good luck!